# [60]Fullerene for Medicinal Purposes, A Purity Criterion towards Regulatory Considerations

**DOI:** 10.3390/ma12162571

**Published:** 2019-08-12

**Authors:** Sanaz Keykhosravi, Ivo B. Rietveld, Diana Couto, Josep Lluis Tamarit, Maria Barrio, René Céolin, Fathi Moussa

**Affiliations:** 1LETIAM, EA7357, IUT d’Orsay, Université Paris Sud, Plateau de Moulon, 91400 Orsay, France; 2Faculté de Pharmacie, Université Paris Descartes, Université Sorbonne Paris Cité, 4 Avenue de l’Observatoire, 75006 Paris, France and Normandie Université, Laboratoire SMS—EA 3233, Université de Rouen, 76821 Mont Saint Aignan, France; 3Grup de Caracterització de Materials, Departament de Física and Barcelona Research Center in Multiscale Science and Engineering, Universitat Politènica de Catalunya, EEBE, Campus Diagonal-Besòs, Av. Eduard Maristany 10-14, 08019 Barcelona, Catalonia, Spain

**Keywords:** fullerene, C_60_, purity, differential scanning calorimetry (DSC), gas chromatography mass spectrometry (GC-MS), nanomaterials, nanomedicine

## Abstract

Since the early nineties countless publications have reported promising medicinal applications for [60]fullerene (C_60_) related to its unparalleled affinity towards free radicals. Yet, until now no officially approved C_60_-based drug has reached the market, notably because of the alleged dangers of C_60_. Nevertheless, since the publication of the effects of C_60_ on the lifespan of rodents, a myriad of companies started selling C_60_ worldwide for human consumption without any approved clinical trial. Nowadays, several independent teams have confirmed the safety of pure C_60_ while demonstrating that previously observed toxicity was due to impurities present in the used samples. However, a purity criterion for C_60_ samples is still lacking and there are no regulatory recommendations on this subject. In order to avoid a public health issue and for regulatory considerations, a quality-testing strategy is urgently needed. Here we have evaluated several analytical tools to verify the purity of commercially available C_60_ samples. Our data clearly show that differential scanning calorimetry is the best candidate to establish a purity criterion based on the sc-fcc transition of a C_60_ sample (T_onset_ ≥ 258 K, ∆_sc-fcc_H ≥ 8 J g^−1^).

## 1. Introduction

Since the early nineties of the last century [60]fullerene (C_60_) [1], the most abundant fullerene, has attracted intense interest in the field of biomedical applications [2,3,4,5,6] due to its unique properties. Dozens of well-characterized C_60_-derivatives have been synthetized and proposed for many biomedical applications [2,3,4,5,6,7,8,9,10]. The first review on this subject published more than 20 years ago already depicted all the potential biomedical applications, thus predicting that fullerenes would lead to a large panel of new drug candidates [6].

One of the most investigated properties in the field of biomedical applications is the unparalleled affinity of C_60_ for radicals [7,8]. The reported activities for C_60_ and its derivatives during the last two decades range from anti-inflammatory and anti-allergic effects to protection against ionizing-radiation-induced mortality and neuroprotection, through limitation of acne vulgaris and even the potentiation of hair growth [2,3]. Most of these beneficial effects are related to its ability to scavenge free radicals. Indeed, several independent teams from different countries, using a large panel of experimental models, have demonstrated that various synthetic compounds with the C_60_ moiety in common can promote and sustain good health [2,3,4,5,6,7,8,9,10] whilst pristine C_60_ itself could extend the expected lifespan of mammals [11]. What, then, prevents all these promising applications from becoming reality [12]? What prevents legal C_60_-based drugs? In fact, the main obstacle is concern about fullerene toxicity raised by some studies conducted with contaminated fullerene samples (see for instance the commentary in Appendix A). Indeed, it has been clearly shown that tetrahydrofuran decomposition products are responsible for the observed toxicity of C_60_-samples, when tetrahydrofuran is used as a co-solvent to prepare C_60_ aqueous suspensions [13]. Two years later another independent research team confirmed that pure C_60_ is not toxic [14]. Meanwhile, a series of tests conducted by several independent teams have confirmed the safety of pure C_60_ in a large panel of experimental models encompassing different clades [13,14,15].

Nevertheless, an officially certified toxicity test is still lacking. Also, C_60_ aggregates may convey toxic elements as any other aggregated material [13,14,15]. Yet, since the publication of the effects of C_60_ on the lifespan of rats [11], a myriad of start-ups has started selling C_60_ oily solutions on a worldwide scale for human consumption without any existing formal oversight [16].

In order to avoid any new toxicity alerts like the ones that occurred in 2004 [17], which would definitively condemn the use of such a promising material in the biomedical realm, a quality-testing strategy is urgently needed. To this purpose, it is of importance to define a purity criterion for C_60_ and to develop simple tools to check the purity of C_60_ samples. Although it is obvious for drug candidates to fulfill a certain set of quality requirements, a purity assessment also falls squarely in the realm of R.E.A.C.H. (Registration, Evaluation, Authorization and restriction of Chemicals), a European regulation “adopted to improve the protection of human health and the environment from risks that can be posed by chemicals” [18].

In this study, we used several methods to assess C_60_ purity both in the solid state and in solution. Solid-state investigations include electron microscopy (SEM), high-resolution X-ray powder diffraction (HR-XRD), differential scanning calorimetry (DSC), thermogravimetric analysis (TGA), and Fourier-transform infrared spectroscopy (FTIR). To assess the presence of impurities through solution and the gas phase, we used reversed-phase high-pressure liquid chromatography (RP-HPLC), gas chromatography coupled to mass spectrometry (GC-MS), and UV–Vis spectrophotometry. The combination of all these techniques was applied to compare the purity of four C_60_ samples from different origins.

The main objective of this work was to select the best and easiest method(s) to verify the purity of a C_60_ sample with the aim of establishing a validation tool for C_60_ used as an active pharmaceutical ingredient (API).

## 2. Materials and Methods

### 2.1. Chemicals

Commercial C_60_ samples with different degrees of purity as specified by the manufacturers: S1 (99.98%); S2 (>99.98%); S3 (99.9%); and S4 (99.5%) were used for this study.

Toluene was obtained from Merck (Darmstadt, Germany) and acetonitrile from Carlo Erba (Val-de-Reuil, France). Both solvents were of high purity grade (99.9%) and were used without further purification.

### 2.2. Scanning Electron Microscopy

Electron microscopy images were obtained using a Hitachi TM-1000 (Tokyo, Japan) tabletop scanning electron microscope with a scanning voltage of 15 kV. To improve the quality of C_60_ sample images, their surface was coated with a thin layer of gold.

### 2.3. High Resolution X-Ray Powder Diffraction

X-ray powder diffraction was carried out on a transmission mode diffractometer using Debye-Scherrer geometry equipped with cylindrical position sensitive detectors (CPS120) from INEL (Artenay, France) containing 4096 channels (0.029° 2θ angular step) with monochromatic Cu Kα_1_ (λ = 1.5406 Å) radiation. Unground specimens were introduced in a Lindemann capillary (0.5 mm diameter) rotating perpendicularly to the X-ray beam during the experiments to improve the average over the crystallite orientations. All samples were measured at 300 K for at least one hour. External calibration using the Na_2_Ca_2_Al_2_F_14_ cubic phase mixed with silver behenate was performed by means of cubic spline fittings. The XRD patterns data were analyzed with FullProf v2.05.18 [19].

### 2.4. Energy Dispersive X-Ray Spectroscopy (EDS) and Energy Dispersive X-Ray Fluorescence (EDF)

Energy dispersive X-ray spectroscopy (EDS) was carried out with a JEOL JSM-7001F field emission scanning electron microscope (Tokyo, Japan) and an X-Max SDD (silicon drift detector) from Oxford Instruments (Abingdon, UK).

Energy dispersive X-ray fluorescence (EDF) was carried out in a helium atmosphere with a Panalytical Epsilon 3XL (Royston, U.K.) equipped with an X-ray tube containing an Ag anode. The C_60_ powder was in the first instance deposited on a Prolene membrane. For quantitative measurements, a suspension of 0.955 g L^−1^ of C_60_ in demineralized water was deposited in quantities of 20 µL (representing 19.1 µg of C_60_) on a polycarbonate membrane. The measurement was carried out after complete evaporation of the water. A control containing only demineralized water was analyzed too.

### 2.5. Differential Scanning Calorimetry (DSC)

DSC measurements at normal pressure were performed by means of a Q100 thermal analyzer from TA Instruments (New Castle, DE, USA) under He flux at cooling then heating rates of 5 K min^−1^ in the temperature range from 303 K to 213 K. Sample masses (≈5 mg) were weighed with a microbalance sensitive to 0.01 mg and sealed in aluminum pans of 30 mL inner volume. Indium was used as a standard for the calibration of temperature and enthalpy change.

### 2.6. Thermogravimetric Analysis

Thermogravimetric measurements were performed by means of a Q50 system from TA Instruments (New Castle, DE, USA) under nitrogen flux from room temperature to 673 K. Heating rates of 10 K·min^−1^ and sample masses of ca. 5 mg were used.

### 2.7. Fourier Transform Infrared Spectroscopy (FTIR)

FTIR analyses were performed on tablet transmission sampling mode. Tablets were prepared by mixing ~1 mg of C_60_ sample powder with 10 mg of KBr powder and then analyzed in a PerkinElmer Frontier IR/FIR (Perkin ELMER, Villebon-sur-Yvette, France) instrument in 400–4000 cm^−1^ ranges.

### 2.8. Reversed-Phase High-Pressure Liquid Chromatography Coupled to Diode Array Detection

Chromatographic analyses were performed as previously described [20] with minor modifications. We used a longer column (25 cm instead of 12.5 cm) packed with the same stationary phase in order to better separate possible impurities eluting before C_60_. The mobile phase consisted of a mixture of toluene/acetonitrile (42/58, *v*/*v*) and was delivered by a P100 Spectra-System pump connected to a UV6000 LP diode array detector (Thermo Spectra-System LC, Les Ulis, France). All chromatographic analyses were performed at 30 °C with a flow rate of 0.8 mL/min.

Working solutions of 1 mg/L, 2 mg/L, 5 mg/L, and 45 mg/L were prepared daily by appropriate dilution of the stock solution (100 mg/L) in the mobile phase before injection of 20 µL into the HPLC system (*n* = 6 injections for each solution and every day for 6 days).

### 2.9. UV-Vis Spectrophotometry

For the UV-Vis spectrophotometry, stock solutions (100 mg/L) were prepared by dissolving each C_60_ sample in toluene. After stirring for 3 days, working solutions of 8 mg/L and 50 mg/L were prepared daily after appropriate dilution in toluene and scanned (300 nm to 700 nm) by an Agilent Varian 50 Bio spectrophotometer (Santa Clara, CA, USA). Direct detection was recorded at 336 nm and 407 nm.

### 2.10. Gas Chromatography Coupled to Mass Spectrometry

Two measurement series were carried out with GC-MS, using the headspace trap (HS) method (Perkin ELMER, Villebon-sur-Yvette, France). C_60_ impurities were analyzed by an HS trap coupled to a Clarus 680 GC/MS platform equipped with a 30 m × 0.25 mm × 0.25 µm Elite-5MS column (Perkin ELMER, Villebon-sur-Yvette, France). C_60_ powder was placed in a closed vial, which was loaded into a thermostatic oven and heated at 160 °C for 20 min. A needle was then lowered into the headspace of the vial, which was pressurized with He as a carrier gas at 35 psi for 1.5 min. While the needle was still in the vial, the carrier flow to the vial was interrupted and by lack of external pressure, the already pressurized headspace vapor was forced to escape through the needle and the trap at 40 °C, where the volatile compounds were collected and focused. This was repeated 4 times, removing practically the entire volatile compounds from the sample vial. The trap was then heated to 280 °C at 20 psi sweeping the desorbed analytes out of the trap and into the Elite-5MS column at 10 psi for GC analysis. For GC separation, the oven was heated from 40 °C to 220 °C with a heating rate of 4 °C min^−1^. The Clarus MS was controlled via TurboMass™5.1 GC/MS software and operated in electron ionization (EI) mode (transfer-line temp: 200 °C, electron energy: 70 eV, detector voltage: 400 V, mass range: 30–300 amu, scan time: 0.41 sec, inter-scan delay: 0.01 sec).

### 2.11. Statistical Analysis

All statistical analyses were performed using one-way ANOVA test with a 95% confidence interval (Graphpad prism 7, Graphpad software, San Diego, CA, USA), and a *p*-value of 0.05 (or less) was considered statistically significant (for at least *n* = 6).

## 3. Results

### 3.1. Solid-State Studies

#### 3.1.1. Morphology of The Samples

The four investigated C_60_ samples—S1, S2, S3, and S4—consist of fine powders exhibiting dark grey to black color for the S1, S2, and S4 samples or a clear brownish hue for the S3 sample. In order to further investigate their crystallinity we examined the samples by electron microscopy.

SEM photographs (Figure 1) show that the C_60_ particles of all the samples exhibit the usual shapes [21,22,23] of face-centered cubic C_60_ microcrystals (i.e., twinned cuboctahedra with well-developed (111) faces while the (100) faces are mostly absent).

Closer analysis revealed that samples S1 and S2 consist of grains of nearly uniform size of about 150 µm in diameter. For samples S3 and S4, the size distribution is much more polydisperse containing grains of various sizes and particles of 1 µm or less in diameter adsorbed at the crystal surfaces. In particular, in sample S4 aggregates of small cubic crystallites are observed (Figure 1).

#### 3.1.2. Crystallographic Studies

Figure 2 shows the high-resolution X-ray powder diffraction (XRD) patterns of the four samples obtained at 300 K.

All samples exhibit the same profile as far as the peak positions are concerned, and the difference, if any, consists of small changes in relative intensities probably due to preferred orientation of the unground crystallites introduced in the Lindemann capillaries. In addition, it can be observed that sample S3 may contain some amorphous material or very small crystals as a slight amorphous halo can be observed around 20° 2θ (the step observed at 4° 2θ is due to alignment of the diffractometer).

The four diffraction patterns are identical to the theoretical pattern. Moreover, none of these four patterns exhibits the “asymmetric ‘shoulder’ superimposed on the normal (111) Bragg reflection” described previously and frequently ascribed to stacking faults and/or residual impurities by several authors [24,25,26].

The lattice parameter of fcc C_60_ at 300 K has been refined for each sample using the pattern matching tool of the FullProf program [19]. It led to the following values: S1: 14.1747(9) Å, S2: 14.1829(7) Å, S3: 14.1950(12) Å, and S4: 14.1707(11) Å.

As the level of crystalline impurity observed by this technique can be as low as 1% for an acquisition time of at least one h, large amounts of crystalline impurities in the form of solvates are clearly not present, although small amounts below the detection limit of 1% cannot be excluded. Moreover, comparing the obtained lattice parameters with those of C_60_O reported in the literature (a/Å ≈ 14.15 − 14.21) [27], it is clear that the presence of C_60_ epoxide in the fcc lattice of C_60_ cannot be excluded as a result of the insignificant difference in unit-cell parameters. The presence of some C_60_ epoxides should be investigated by other techniques. Hence, with XRD, it can be confirmed that no significant amount of crystalline impurities was present, however sample S3 contained some amorphous (non-crystalline) material.

#### 3.1.3. Trace Analysis in the Solid State

All four samples were analyzed by EDS. No other elements were found in samples S1, S2, and S3, other than carbon and a trace of oxygen (Appendix A). In sample S4, small particles containing sulfur were observed. To quantify this finding EDF was used. A simple dry deposit analyzed by EDF indicated the presence of sulfur and a trace amount of iron. Using liquid suspensions as described in the experimental section, it was found that the sample contained 0.36% of sulfur and 0.08% of iron. After multiple deposits, the presence of copper and chromium could be observed, but these quantities were below the limits of quantification (Appendix A). It is most likely that the sulfur is a residue of carbon disulfide, in which C_60_ is relatively soluble and which could have been used as a solvent for processing. Iron and the traces of copper and chromium were most likely due to stainless steel equipment used to process the C_60_.

#### 3.1.4. Calorimetric Results

Figure 3 shows the results obtained from heating the four studied samples and the temperature and enthalpy changes with uncertainties of ±1 K and ±5% of the enthalpy value are summarized in Table 1.

Although the melting temperature of fcc C_60_ remains unknown, because it sublimes or decomposes before melting, early DSC studies have shown that a first-order transition into a simple cubic (sc) structure occurs on cooling, which reverts back to the fcc phase at 257.1 K on reheating [28,29]. Using a large single crystal, it was found that the onset temperature of this transition occurs at 261.4 K with an enthalpy change of 9.0 ± 0.5 kJ mol^−1^ (12.5 ± 0.7 J g^−1^) including a 2.2 kJ mol^−1^ (3 J g^−1^) pre-transitional effect starting at about 220 K. If the pre-transitional effect is not considered, the enthalpy change of the endothermic peak at 261.4 K was found to be 6.8 kJ mol^−1^ (9.5 ± 0.7 J g^−1^).

Considering that enthalpy changes and an onset temperature closest to the result for the single crystal are indicative of a higher purity, it can be concluded that samples S1 and S2 clearly have a higher purity than the other two samples S3 and S4. For S3, the decrease in enthalpy fits the XRD observation of the presence of amorphous material, although the lowering of the transition temperature also points to the presence of impurities; whereas in the case of S4, a possible presence of impurities, such as the sulfur observed by EDS and EDF, may explain the lowering of the value of the transition enthalpy.

#### 3.1.5. Thermogravimetric Results

Thermogravimetric measurements were run from room temperature to 673 K. C_60_ samples were placed in pierced aluminum pans and an empty pan was used as a control. It can be seen (Figure 4) that no significant weight loss occurs up to about 600 K with S1, S2, and S3. In contrast, sample S4 exhibits a sudden weight loss of about 2% in the 350−450 K range that can be ascribed to molecular materials with a relatively high vapor pressure, which are most likely residual solvents trapped in the fcc lattice. The weight loss of about 1% in the 600−650 K range observed for S1 cannot be explained by the loss of residual solvent, since volatile organic solvents used to extract and purify C_60_, when trapped in the fcc lattice, are known to leave the C_60_ lattice in the temperature range of 300−450 K.

Thus, TGA experiments confirm that samples S1 and S2 possess a high purity. The fact that no mass loss was found for sample S3, which clearly contains impurities following the DSC results, can tentatively be explained by impurities composed of polymer-like C_60_ substances, possibly induced by light exposure or by volatile impurities that leave at the very beginning of the experiment, when the TGA balance is still stabilizing [30]. The presence of volatile impurities is confirmed for sample S4.

#### 3.1.6. Infrared Spectroscopy

To investigate, in particular, the impurities of sample S3 in the solid state, also in relation to its clear brownish hue, FTIR was used. FTIR was used before to compare the purity of C_60_ samples and the presence of additional peaks in C_60_ spectra was reported for certain samples [31,32].

Figure 5 shows the representative spectra of the samples S1 and S3. The peaks at 526 cm^−1^, 575 cm^−1^, 1181 cm^−1^, and 1427 cm^−1^ correspond to C_60_ vibrations (Figure 5A) [33,34].

What is worth noting however is that both samples exhibit additional peaks at 2850–3000 cm^−1^, which can be ascribed to saturated C–H bonds [35], as well as some peaks at 796 cm^−1^, 1044 cm^−1^, and 1259 cm^−1^, which can be ascribed to methyl-siloxane derivatives [36]. Moreover, for sample S3, additional peaks at 669 cm^−1^, 738 cm^−1^, 775 cm^−1^, and 802 cm^−1^ most likely related to aromatic C–H vibrations (Figure 5B) were also observed. These latter peaks closely correspond to those typical for toluene and xylene [35], indicating that despite the absence of weight loss, sample S3 contains small amounts of aromatic solvents, whereas sample S1 does not.

### 3.2. Solution- and Gas-Phase-Based Studies

#### 3.2.1. UV-Vis Spectroscopy

Considering that impurities that interact with C_60_ could affect the molar absorptivity of C_60_-samples, UV-Vis spectroscopy should be considered as one of the simplest tools to check the purity of a C_60_-sample.

Figure 6A shows the representative UV-Vis spectrum of a toluene solution of the studied samples. The observed spectral features agree with those previously observed by other authors [37,38]. The absence of absorption bands at 424 nm and 496 nm, characteristic of C_60_O [39], clearly indicates that this impurity can only be present at concentrations below the limit of detection.

Figure 6C shows the comparison between the calculated molar absorptivity of each sample at 336 nm. The obtained data clearly demonstrates that while there is a significant difference between samples S1 and S3, and S2 and S3, there is no significant difference between sample S4 and the three other samples, probably because of the low concentration of the impurities in the latter. The within-run (*n* = 6) and between-run (*n* = 6) precisions of the method under the selected conditions are 0.1% and 0.5%, respectively. Hence, more precision is needed for UV-Vis comparison of these samples. Further analyses in other solvents with a lower UV cutoff such as hexane or methanol may provide additional information in the UV region [40,41]; however, C_60_ is sparingly soluble in these solvents and therefore high-precision results may be difficult to obtain for the impurities present at a very low concentration.

#### 3.2.2. High-Pressure Liquid Chromatography (HPLC)

HPLC is commonly used to check the purity of C_60_ samples, notably in order to check the presence of higher fullerenes and/or C_60_ derivatives. Hence, we analyzed the samples by HPLC coupled to diode array detection with a previously validated method [20] and a minor modification. Notably, we doubled the column length in order to enhance the resolution, thus improving the separation of possible impurities eluting before C_60_. Such impurities may include peaks corresponding to more polar compounds such as oxide derivatives. However, other aromatic compounds such as benzene and anthracene derivatives cannot be detected under these chromatographic conditions because of the UV cut-off of the toluene-based mobile phase.

Injecting a 5 mg/L C_60_ solution showed no additional peaks, thus indicating the absence of significant amounts of higher fullerenes or C_60_-oxides in the samples. Nevertheless, the injection of an overloading C_60_ concentration (Figure 7A) shows at least six minor peaks (Figure 7A, insert I_1_) eluting before the C_60_ peak. Based on their spectral features (Figure 7A, insert I_2_), these minor peaks can be ascribed to some C_60_-oxides [39].

Figure 7B summarizes the comparison between the area/concentration ratios of each C_60_ sample. With a precision of 5.8% (*n* = 6 obtained over three days), the HPLC analysis shows no significant difference between the samples. Hence, the precision of this technique cannot provide a quantitative comparison between the studied samples. This outcome agrees with those published previously [32].

#### 3.2.3. GC-MS Analyses

As TGA and FTIR data clearly show the presence of volatile impurities, GC-MS was carried out in order to identify these impurities.

Figure 8 shows the obtained GC-MS chromatographic profiles. They clearly show the presence of several volatile impurities namely benzene, naphthalene, phthalate, and siloxane derivatives (Table 2). These semi-quantitative analyses confirm the data obtained by the analyses in the solid state indicating that the samples can be classified as a function of their purities as follow: S1 > S2 > S4 > S3. Notably, the S3 and S4 samples contain significant amounts of xylene and toluene and some naphthalene derivatives as compared to the S1 and S2 samples.

These data show that the GC-MS method can be used after appropriate calibration and validation to quantify the residual solvents and other volatile impurities in a C_60_ sample.

## 4. Discussion

In order to perform successful clinical trials, the purity of an API must be guaranteed and so does the purity of C_60_, which is already being sold for human consumption even if no official certified procedure exists to verify its purity [16].

This paper contains a comprehensive analytical study of C_60_ samples, which has never been carried out before. It illustrates the advantages and limitations of the techniques to characterize and evaluate the purity of C_60_ samples.

While the physicochemical properties of C_60_ are well-known [7,10,22,23,24,25,26,27], until now only three studies have been devoted to the purity analysis of C_60_ samples [31,32,42]. The first study conducted more than 25 years ago concluded that C_60_ is prone to absorb solvents and oxygen, which may change the properties of the sample, and that chemical reactions can occur with absorbed species even in the dark [42]. After comparing the performances of several characterization techniques including fast atomic bombardment (FAB—mass spectrometry, FTIR, UV-VIS spectrometry, and scanning electron microscopy (SEM), the second study concluded that thermal analysis is a powerful technique for analyzing the purity of C_60_ samples [31]. The third report, published eleven years later, concluded that chromatographic analysis alone, as currently used in the certificates of C_60_ samples, is not sufficient for detecting all impurities. According to the authors, it is necessary to use additional methods such as mass spectrometry [32]. However, a straightforward purity criterion for C_60_ samples is still lacking and no regulatory recommendations exist up to now.

The obtained data overall agree with previous studies [31,32,42]. They further show that DSC is the most appropriate technique to establish a rapid and simple purity criterion for C_60_ samples based on the sc-fcc transition temperature and enthalpy change (T_onset_ ≥ 258 K, ∆_sc-fcc_H ≥ 8 J g^−1^).

Considered together, the results of the solid-state studies clearly show that samples S1 and S2 are the purest, whereas samples S3 and S4 contain some impurities. It is however not possible to determine the identity of these impurities by the solid-state analytical methods, which will be carried out with a further series of analytical methods.

FTIR provides a rapid assessment of the kind of impurities present in a C_60_ samples, in particular because C_60_ itself contains only a limited number of FTIR absorption peaks; thus, impurity peaks are quickly recognized.

GC-MS provides the most detailed picture of the various volatile molecular impurities that can be found in the different C_60_ samples, whereas HPLC provides information on the presence of higher fullerenes and oxidized fullerene species. Nonetheless, even the visual aspect of the C_60_ samples can already provide clues about the quality of sample S3, which contains most kinds of impurities and is least crystalline, possesses a brownish color, whereas the highly pure and crystalline samples S1 and S2 have a black metallic appearance.

The next step is the verification of the innocuousness of C_60_ oily solutions and other C_60_-containing preparations proposed by some companies for human consumption. Indeed, C_60_ oily solutions prepared under some conditions may be harmful [43].

## Figures and Tables

**Figure 1 materials-12-02571-f001:**
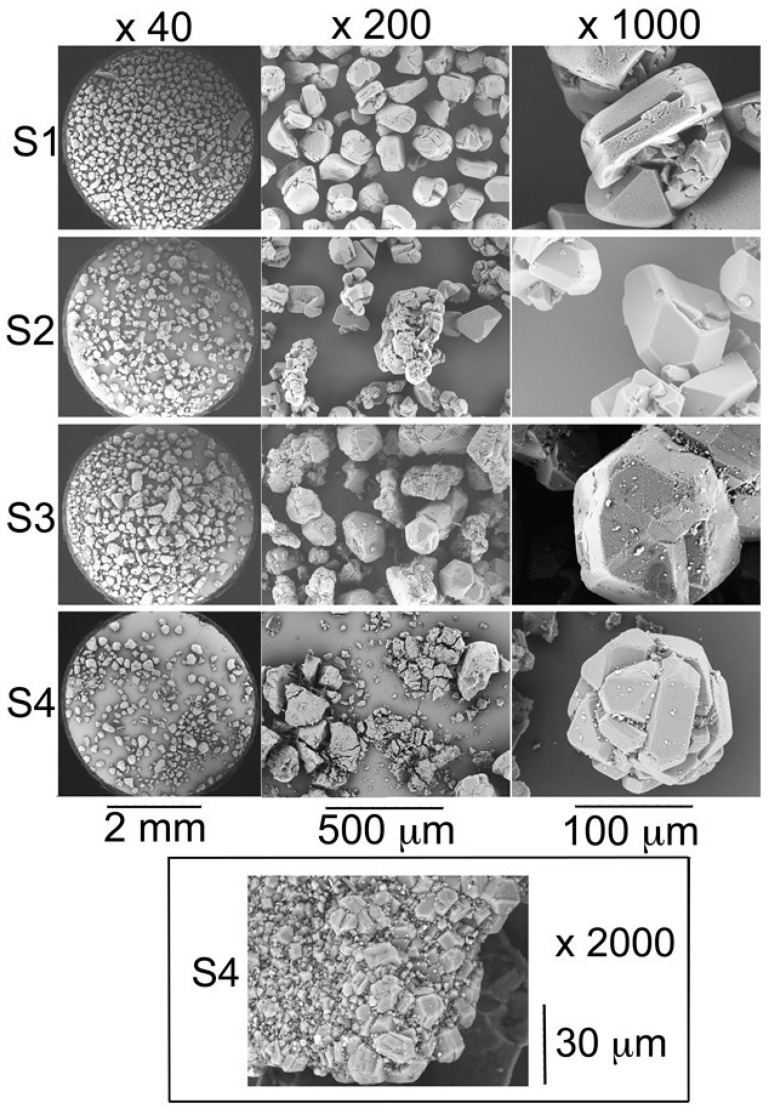
Scanning electron microscopy photographs of crystalline C_60_ powders of samples S1, S2, S3, and S4 with different magnifications. The inset shows the photograph of an aggregate of C_60_ crystals found in sample S4.

**Figure 2 materials-12-02571-f002:**
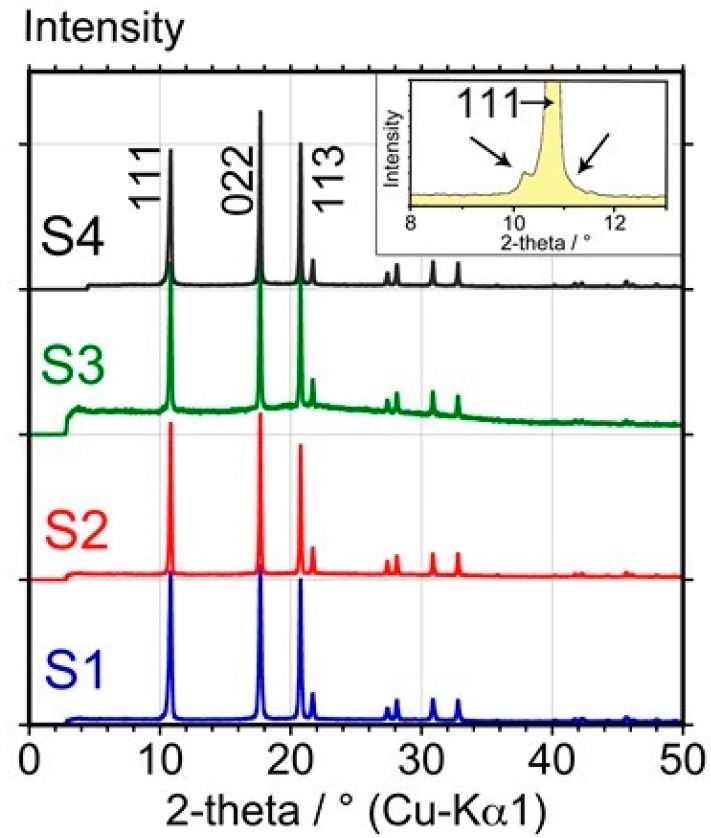
High-resolution X-ray powder diffraction patterns of samples S1, S2, S3, and S4, obtained at 300 K. The arrows in the inset show how stacking faults and impurities modify the profile of the 111 reflection.

**Figure 3 materials-12-02571-f003:**
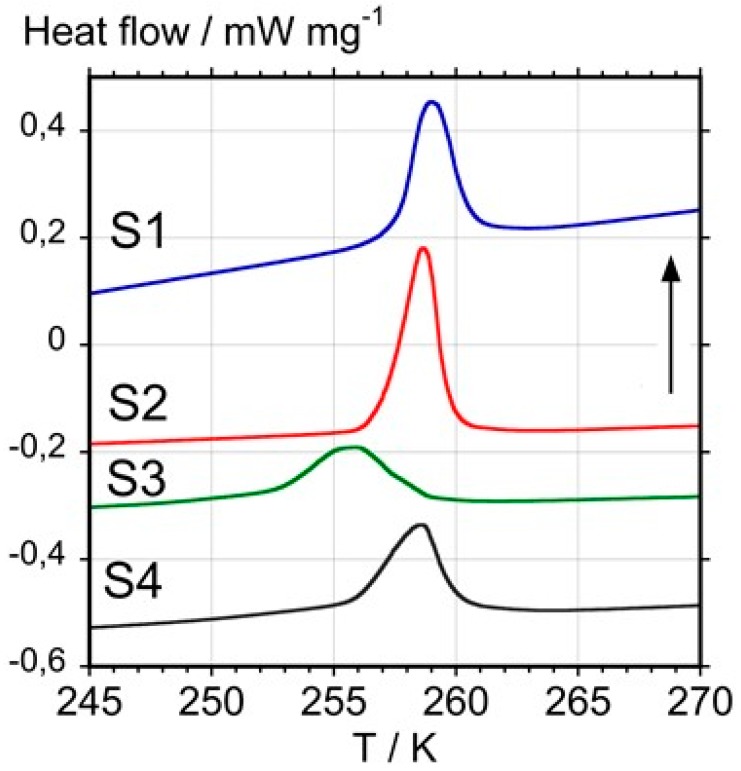
Differential scanning calorimetry curves obtained with samples S1, S2, S3, and S4 at a heating rate of 10 K min^−1^. The vertical arrow indicates the endothermic direction.

**Figure 4 materials-12-02571-f004:**
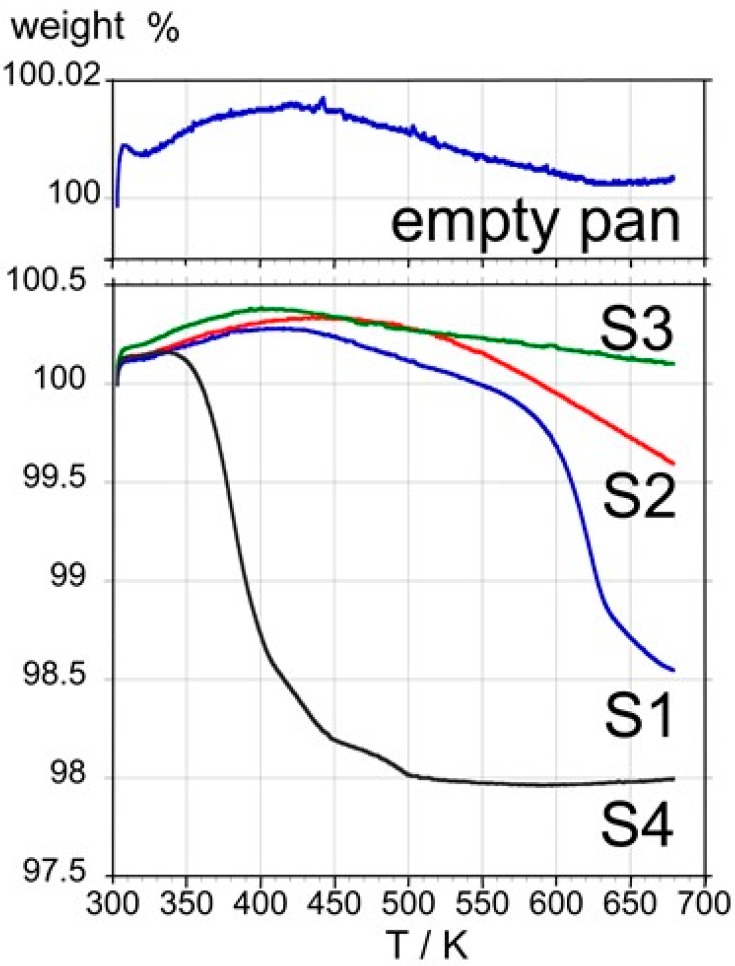
Thermogravimetric curves obtained with samples S1, S2, S3, S4, and an empty pan at a heating rate of 10 K min^−1^.

**Figure 5 materials-12-02571-f005:**
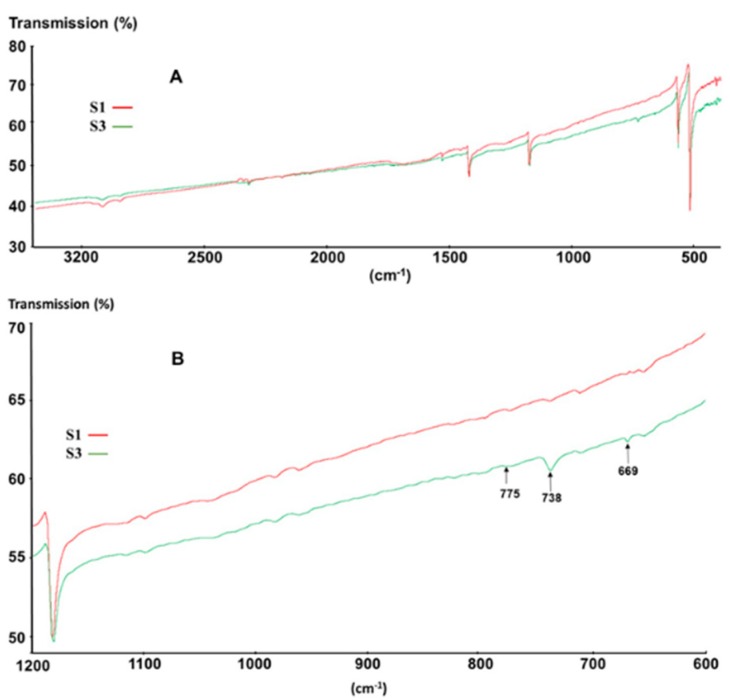
(**A**) FTIR (Fourier-transform infrared spectroscopy) spectra of S1 (red) and S3 (green); (**B**) magnification of (A).

**Figure 6 materials-12-02571-f006:**
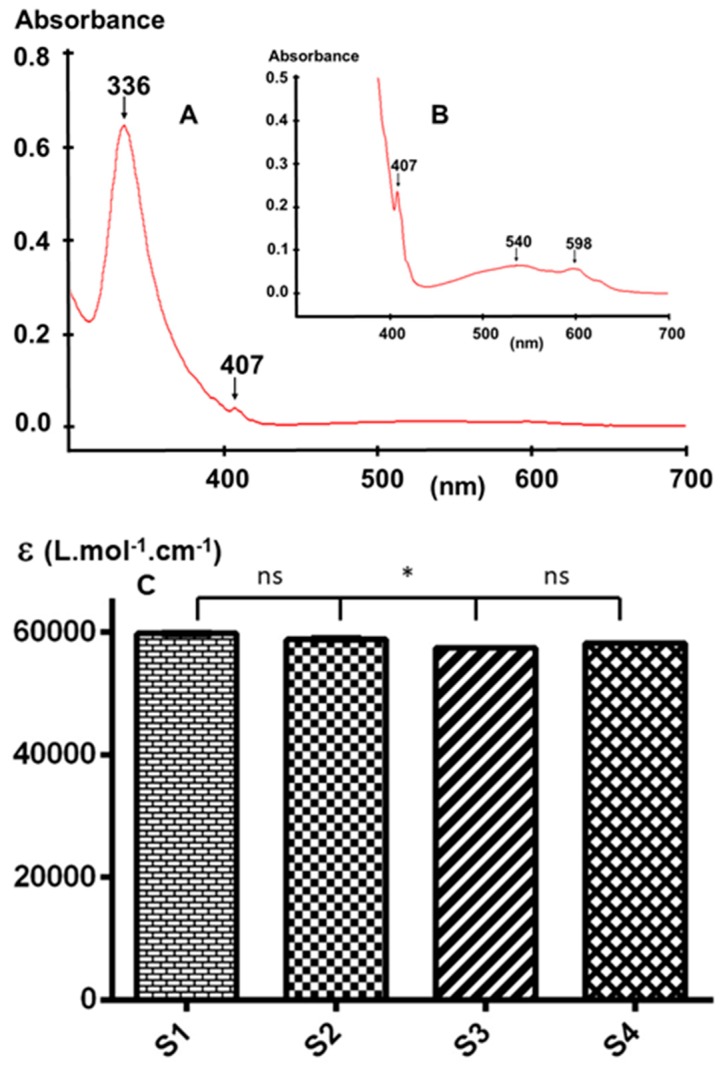
Representative UV-Visible spectrum of (**A**) a C_60_ sample; (**B**) magnification of (A); and (**C**) comparison between the molar absorptivity of the four samples. (ns = not significant).

**Figure 7 materials-12-02571-f007:**
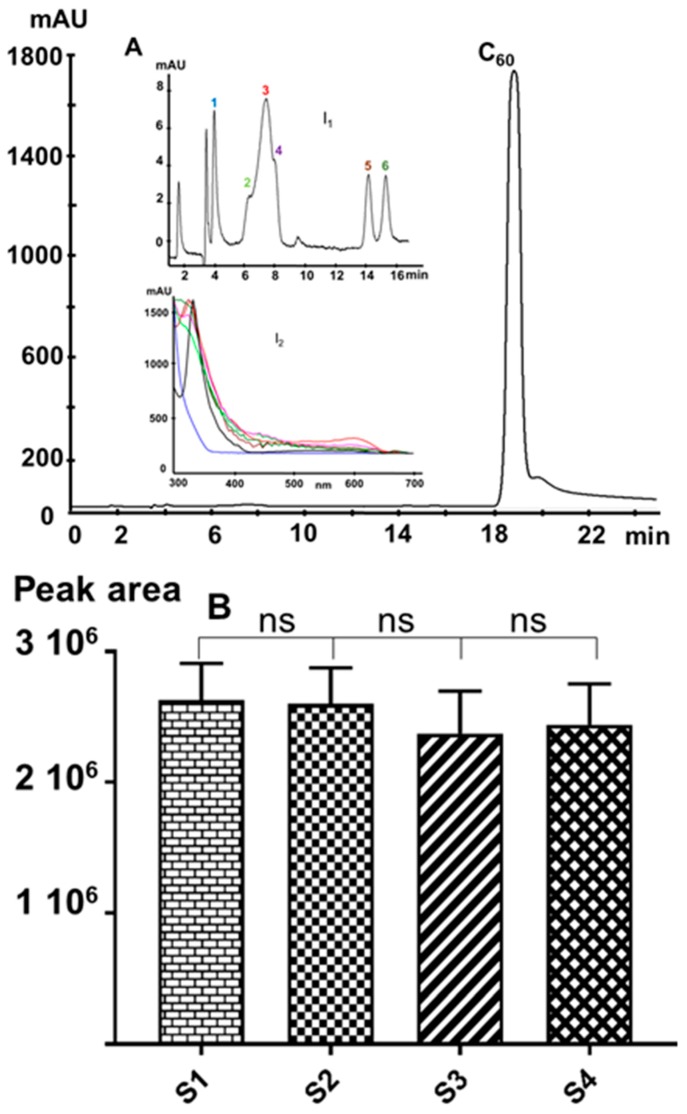
Representative reversed-phase (RP)-chromatographic profile of a C_60_ sample: (**A**) 45 mg/L (the shoulder and the tailing peak are due to mass overload), (I_1_) magnification of (A), and (I_2_) extracted spectra of 1 to 6 peaks; (**B**) comparison between the peak area/C_60_-concentration ratios of the four samples injected at a concentration of 5 mg/L. (ns = not significant).

**Figure 8 materials-12-02571-f008:**
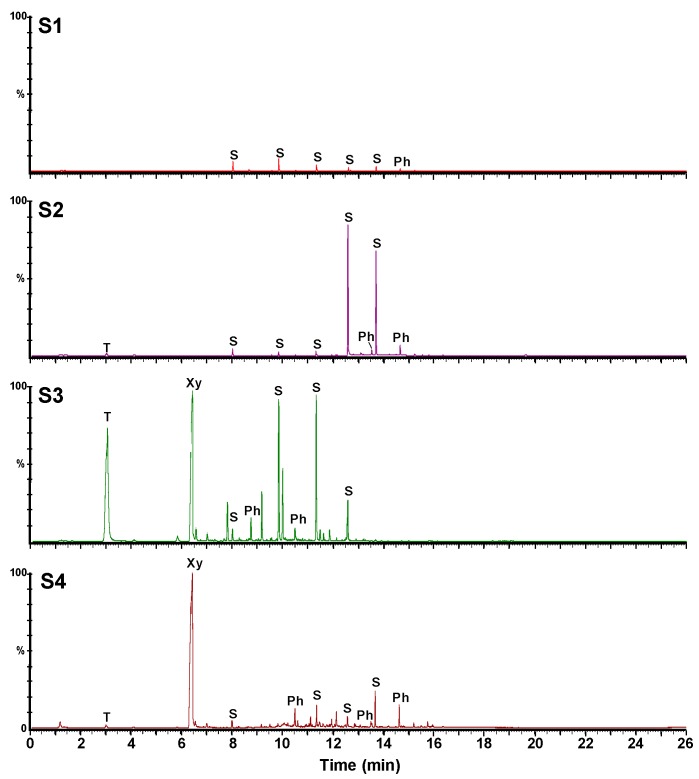
GC-MS (gas chromatography–mass spectrometry) chromatograms of the four C_60_ samples (T: toluene, Xy: xylene, S: siloxane derivatives, Ph: phthalate derivatives).

**Table 1 materials-12-02571-t001:** Temperature and enthalpy changes.

Samples	T_onset_/°C (K)	ΔH/J g^−1^
S1	−15.1 (258.0)	8.4
S2	−13.1 (260.0)	9.2
S3	−18.5 (254.6)	6.8
S4	−15.9 (257.2)	6.9

**Table 2 materials-12-02571-t002:** Detected impurities in the four C_60_ samples (semi-quantitative aspect; − absence; and + presence).

Impurity	S1	S2	S3	S4
Toluene	−	+	+++	+
Xylene	−	−	+++	+++
Trimethyl benzene	−	−	±	−
Tetra methyl benzene	−	+	+++	−
Penta methyl benzene	−	−	+	−
Hexa methyl benzene	−	−	+	−
Hydroxylated toluene	−	−	−	+
Dichlorobenzene	−	−	±	−
Methylbenzaldehyde	−	−	+	−
Hydroxymethylbenzoic acid	−	−	+	−
Naphthalene	−	−	+	−
Methylnaphthalene	−	−	+++	++
Dimethylnaphthalene	−	−	+	+
Trimethylnaphthalene	−	−	+	−
Tetramethylbiphenyl	−	−	−	+
Dibuthyl Phtalate	±	+	±	+++
Phthalic acid, isobuthyl octadecyl ester	++	+++	±	+++
Diethyl Phthalate	±	++	±	++
Benzoic acid, 4-ethoxy-, ethyl ester	−	++	−	++
Octamethylcyclotetrasiloxane	++	−	++	−
Decamethylcyclopentasiloxane	++	+	+++	+
Dodecamethylcyclohexasiloxane	+	+	+++	++
Tetradecamethylcycloheptasiloxane	+	+++	+++	++
Hexadecamethylcyclooctasiloxane	+	+++	±	+++
Octadecamethylcyclononasiloxane	+	++	−	++

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
