# Peer review of "[60]Fullerene for Medicinal Purposes, A Purity Criterion towards Regulatory Considerations"

_materials, 2019, doi:10.3390/ma12162571_

Round 1

Reviewer 1 Report

The Authors have submitted a manuscript on the relative toxicity of C60 fullerene. 

The manuscript is interesting and its goal is relevant to the current state of biomedical research. Overall, I support its acceptance after the following minors will be addressed in order to further improve its impact:

-       Authors do not report anything about Good Laboratory Practice (GLP). I think that in this paper they should explain what GLP is and how can help to fill the purity issue of C60 fullerene.

-       Even if Authors do not work under the GLP regulations, they should describe in detail all reported analysis and state about their standardization. 

-       There are similar issues on CTAB for gold nanorods and in general on persistence of noble metals. Authors should report a sentence to let the Readers knows this is a common and general problem for nanomaterials. They can refer to both, doi:10.1007/s11051-010-9911-8 and 10.1021/acs.bioconjchem.7b00664

-       Authors should report some statements regarding the use of animals (such as zebrafish) for toxicity evaluation. They can refer to, doi: 10.1016/j.carbon.2007.04.021

Author Response

Reviewer 1.

The Authors have submitted a manuscript on the relative toxicity of C60 fullerene. 

The manuscript is interesting and its goal is relevant to the current state of biomedical research. Overall, I support its acceptance after the following minors will be addressed in order to further improve its impact:

-       Authors do not report anything about Good Laboratory Practice (GLP). I think that in this paper they should explain what GLP is and how can help to fill the purity issue of C60 fullerene.

-       Even if Authors do not work under the GLP regulations, they should describe in detail all reported analysis and state about their standardization. 

-       There are similar issues on CTAB for gold nanorods and in general on persistence of noble metals. Authors should report a sentence to let the Readers knows this is a common and general problem for nanomaterials. They can refer to both, doi:10.1007/s11051-010-9911-8 and 10.1021/acs.bioconjchem.7b00664

-       Authors should report some statements regarding the use of animals (such as zebrafish) for toxicity evaluation. They can refer to, doi: 10.1016/j.carbon.2007.04.021

Response

We sincerely thank the reviewer for his valuable comments concerning GLP, common and general problem for nanomaterials, and the use of zebrafish for toxicity evaluation. Nonetheless, we think that the main objective of our work is to help establish a purity criterion for [60] fullerene samples, an issue that should be dealt with even before toxicity or GLP regulations can be considered. We are therefore very far from these considerations for the moment. As soon as we look at the problem of clinical trials in our future work, we will not fail to refer to all these valuable references indicated by the reviewer, of course.

Reviewer 2 Report

The paper of Moussa et al. is interesting and face topics too often neglected: 1) the definition of a purity criterion for C60; 2) the development of simple tools to check the purity of C60 samples.

So, this work may deserve the publication in Materials, but I have a very important question that prevent the acceptance of the paper in the current form:

What is the statistical significance of the experiments?

I want to see, at least for one sample, three replicates and three repeats, using the different methodologies showed in the paper, to understand the statistical significance of the experiments.

In addition, please indicate clearly (seller and identification code) the origin of the C60 samples, these data are crucial for the validation of the experiments.

Author Response

Comments and Suggestions for Authors

The paper of Moussa et al. is interesting and face topics too often neglected: 1) the definition of a purity criterion for C60; 2) the development of simple tools to check the purity of C60 samples.

So, this work may deserve the publication in Materials, but I have a very important question that prevent the acceptance of the paper in the current form:

What is the statistical significance of the experiments?

I want to see, at least for one sample, three replicates and three repeats, using the different methodologies showed in the paper, to understand the statistical significance of the experiments.

Response:

First of all we sincerely thank the reviewer for accepting to review our paper.

It is our pleasure to make available to the reviewer the tables of the values obtained for the UV-Visible spectroscopy and the HPLC. As we used “Graphpad Prism Â» for statistics and therefore we have also reported the data in a ppt file, in case the reviewer does not have access to the software.

In addition, please indicate clearly (seller and identification code) the origin of the C60 samples, these data are crucial for the validation of the experiments.

Response:

We understand the point of view of the reviewer as revealing the origin of the samples is in most cases essential for qualified and quantified comparison of scientific results. We sincerely thank him for raising this issue.

In fact, we have given this point a lot of thought and we have decided not to publish the origin of the samples for several reasons:

1.       The main objective of this work is to define a purity criterion for a C60 sample. Our work should therefore be sample independent and it should be possible to validate our paper with any C60 sample available on the market. The purity criterion should allow a skilled person to check the purity of any C60 sample.

2.       We used 4 samples of different origins in order to illustrate and validate the analytical approach, because a single sample of a single origin would not be statistically significant.

3.       More importantly, it cannot be our role to classify specific samples according to their origin and purity as we would in no way want to judge the quality of the manufacturer. This is simply not the objective of this work.

We are of course ready to make these samples available to anyone wishing to verify the data we have obtained.

Reviewer 3 Report

Moussa and co-worker present in their submission to Materials "[60]Fullerene for medicinal purposes, a purity criterion towards regulatory considerations". This is a submission with many issues, which all must be carefully addressed by the authors. A critical re-evaluation will tell, whether the revision can be accepted or not.

Abstract: "C60 samples labeled as pure still contain several impurities" This is a very problematic statement that must be revised, if not deleted. As written in subchapter 2.1, all samples have a C60 content of lower than 100% and therefore all MUST contain impurities. Furthermore, the authors do not show (other than in the abstract), which producer declared their samples as "pure" (if they did it at all). The same comments apply to the following statement "Meanwhile, a series of tests conducted by several independent teams have confirmed the safety of pure C60 in a large panel of experimental models encompassing different clades [14–16]."

"In fact, the main obstacle is concern about fullerene toxicity raised by some studies conducted with contaminated fullerene samples [13]." It must be stated, which impurities were thought to be connected with the toxicity of fullerene, as ref. 13 is not easily available.

"Finally, as XRD analysis did not evidence metals in the samples, we decided not to investigate further these kind of impurities" This statement is again highly problematic, as XRD would only indicate metal impurities, if a certain metal would be present at a concentration of higher than 1% and in a crystalline form. Generally speaking, XRD is NOT the way to determine general metal impurities in APIs.

The journal names must be correctly abbreviated/written. This is not the case for references 6, 8, 10 and 40.

For reference 12 the article number rather than the non-informative page numbers should be given.

The page number in reference 16 are wrong, the editors are missing.

The page number and the editors in reference 34 are missing.

The title and the journal title of reference 36 are incorrectly written.

The journal name of ref. 37 is incorrectly written, additional the volume and the page numbers are wrong.

The journal name of ref. 43 is incorrectly written, additional the issue number must be deleted.

Author Response

Reviewer 3

Comments and Suggestions for Authors

Moussa and co-worker present in their submission to Materials "[60]Fullerene for medicinal purposes, a purity criterion towards regulatory considerations". This is a submission with many issues, which all must be carefully addressed by the authors. A critical re-evaluation will tell, whether the revision can be accepted or not.

Abstract: "C60 samples labeled as pure still contain several impurities" This is a very problematic statement that must be revised, if not deleted. As written in subchapter 2.1, all samples have a C60 content of lower than 100% and therefore all MUST contain impurities. Furthermore, the authors do not show (other than in the abstract), which producer declared their samples as "pure" (if they did it at all).

Response:

First of all, we sincerely thank the reviewer for accepting to review this paper. Also, we must thank him for his valuable comment on the sentence “"C60 samples labeled as pure still contain several impurities". In fact, we used four samples including one sample with 99.98 % purity, one sample containing sublimed C60 (99.99 % purity), and two samples with 99.95 % purity both labelled as highly pure by the sellers, which may be misleading for the buyers depending on the intended use. Therefore, the reviewer is right and the sentence we wrote in the abstract may be misleading as all these samples MUST contain impurities, as noticed by the reviewer. Hence we decided to delete this sentence according to the suggestion of the reviewer in the revised version.

The same comments apply to the following statement "Meanwhile, a series of tests conducted by several independent teams have confirmed the safety of pure C60 in a large panel of experimental models encompassing different clades [14–16]."

Response:

This statement is strongly supported by numerous papers that we cannot quote here in their entirety. However, the quoted references (14-16) perfectly support this statement, notably the reference 16 is a general review on this topic that exhaustively comments nearly all the papers on fullerene toxicity including those reporting negative effects.

"In fact, the main obstacle is concern about fullerene toxicity raised by some studies conducted with contaminated fullerene samples [13]." It must be stated, which impurities were thought to be connected with the toxicity of fullerene, as ref. 13 is not easily available.

Response:

The reviewer is right, reference 13 alone is not sufficient to support the statement. Hence, we have added the ref. 14 to 16 in the revised version, where Henry et al. [14] clearly show that tetrahydrofuran decomposition products are responsible of the observed toxicity of fullerene. Indeed, the authors who observed the toxic effects used tetrahydrofuran as co-solvent to prepare the C60 aqueous suspensions. Moreover, Spohn et al. [15] confirmed that C60 devoid of any apparent impurities is not toxic while ref [16] is a general review thoroughly commenting the essential papers on the subject at the time. We modified the text in the revised version accordingly.

"Finally, as XRD analysis did not evidence metals in the samples, we decided not to investigate further these kind of impurities" This statement is again highly problematic, as XRD would only indicate metal impurities, if a certain metal would be present at a concentration of higher than 1% and in a crystalline form. Generally speaking, XRD is NOT the way to determine general metal impurities in APIs.

Response:

Although metals can be observed by X-ray diffraction, the reviewer is right that XRD does not observe trace impurities of metals. As we did not observe the presence of metallic crystals in the examined samples and as we know by experience that C60 samples of 99.9 % of purity will not contain metallic impurities due to the way C60 is produced and interacts with metals, we decided not to investigate this further. Moreover, the presence of metals, whether we have observed them analytically or not, would be picked up in the overall purity assessment that we propose in our paper. In the case that metals are present, the weight of the sample would increase without the C60 transition enthalpy reflecting the proper weight of the sample, in particular because the specific weight of metals are much larger than that of carbon. However, the reviewer is right that our analysis is not entirely complete. We have therefore decided to check the absence of metallic impurities by Energy Dispersive X-ray Spectroscopy (EDS) and Energy Dispersive X-ray Fluorescence (EDF). And, thanks to the reviewer we disclosed that sample 4 contains relatively high amounts of sulfur 0.36 % and 0.08 % of iron. And, after multiple deposits, a presence of copper and chromium could be observed, but these quantities were below the limits of quantification (in figure S2 of the supplementary materials peaks of the respective metals can be observed). It is most likely that the sulfur is a residue of carbon disulfide, in which C60 is relatively soluble and which could have been used as solvent for processing. Iron and the traces of copper and chromium are most likely due to stainless steel equipment used to process the C60. All these results have been added in the revised version with the corresponding supplementary figures.

The journal names must be correctly abbreviated/written. This is not the case for references 6, 8, 10 and 40.

Response:

Corrections have been carried out for all mentioned references.

For reference 12 the article number rather than the non-informative page numbers should be given.

The page number in reference 16 are wrong, the editors are missing.

The page number and the editors in reference 34 are missing.

The title and the journal title of reference 36 are incorrectly written.

The journal name of ref. 37 is incorrectly written, additional the volume and the page numbers are wrong.

The journal name of ref. 43 is incorrectly written, additional the issue number must be deleted.

Round 2

Reviewer 3 Report

Moussa and co-worker have substantially improved their submission to Materials "[60]Fullerene for medicinal purposes, a purity criterion towards regulatory considerations". Therefore the manuscript can be published as it is now.